# Distinct representational properties of cues and contexts shape fear and reversal learning

**Antoine Bouyeure[1]\*, Daniel Pacheco-Estefan[1,2], George Jacob[1], Malte Kobelt[1], Marie-Christin Fellner[1], Jonas Rose[3], Nikolai Axmacher[1]\***

[1]Department of Neuropsychology, Ruhr-Universität Bochum, Bochum, Germany; [2]Department of Basic, Developmental and Educational Psychology, Faculty of Psychology, Autonomous University of Barcelona, Barcelona, Spain; [3]Department of Neuroscience, Ruhr-Universität Bochum, Bochum, China

## eLife Assessment

This is an **important** study with **convincing** evidence that multi-voxel fMRI activity patterns for threat-conditioned stimuli are altered by learning CS-US contingencies. The analyses are dense, but rigorous. The protocol is quite nuanced and complex, but the authors have done a fair job of explaining and presenting the results. The work is relevant for our understanding of how effective learning changes neural stimulus representation in the human brain.

**\*For correspondence:**
antoine.bouyeure@rub.de (AB);
nikolai.axmacher@rub.de (NA)

**Competing interest:** The authors declare that no competing interests exist.

**Abstract** When we learn that something is dangerous, a fear memory is formed. However, this memory is not fixed and can be updated through new experiences, such as learning that the threat is no longer present. This process of updating, known as extinction or reversal learning, is highly dependent on the context in which it occurs. How the brain represents cues, contexts, and their changing threat value remains a major question. Here, we used functional magnetic resonance imaging and a novel fear learning paradigm to track the neural representations of stimuli across fear acquisition, reversal, and test phases. We found that initial fear learning creates generalized neural representations for all threatening cues in the brain's fear network. During reversal learning, when threat contingencies switched for some of the cues, two distinct representational strategies were observed. On the one hand, we still identified generalized patterns for currently threatening cues, whereas on the other hand, we observed highly stable representations of individual cues (i.e. item-specific) that changed their valence, particularly in the precuneus and prefrontal cortex. Furthermore, we observed that the brain represents contexts more distinctly during reversal learning. Furthermore, additional exploratory analyses showed that the degree of this context specificity in the prefrontal cortex predicted the subsequent return of fear, providing a potential neural mechanism for fear renewal. Our findings reveal that the brain uses a flexible combination of generalized and specific representations to adapt to a changing world, shedding new light on the mechanisms that support cognitive flexibility and the treatment of anxiety disorders via exposure therapy.

## Introduction

Fear acquisition describes the process by which a previously neutral cue, the conditioned stimulus (CS), becomes associated with an aversive, unconditioned stimulus (US), and eventually comes to evoke fear on its own. It is typically a rapid and robust process that may create long-lasting fear memories that can persist after the threats have passed. This persistence can be evolutionarily advantageous,

since it may be adaptive to respond to a false alarm rather than miss a potential threat. However, failing to suppress a fear response in the absence of actual danger can be maladaptive and has been proposed as a key etiological factor in conditions, such as anxiety disorders and post-traumatic stress disorder (*Milad and Quirk, 2012*).

While fear acquisition is rapid and robust, the suppression of fear responses in the absence of the US – i.e., fear extinction – is strongly context-dependent and more flexible (*Maren et al., 2013*; *Milad and Quirk, 2012*; *Liu et al., 2024*). This is demonstrated by the phenomena of spontaneous recovery, renewal, and reinstatement, all of which show that the original fear memory can resurface under certain conditions (*Bouton, 2002*). Specifically, fear renewal reflects a return of fear responses after a change in context, showing that extinction does not erase the original fear memory trace but inhibits it selectively within the extinction context (*Greco and Liberzon, 2016*). Learning and extinction do not occur only in relation to fear but also during reinforcement learning and reversal, reflecting sensitivity to contingency changes more generally (*Schiller et al., 2008*; *Wisniewski et al., 2023*). In these cases, the context dependency of extinction may support cognitive flexibility since appropriate actions can be selected according to situational demands (*Schiller and Delgado, 2010*; *Chaby et al., 2019*; *Xin et al., 2024*). Contrastingly, the context specificity of extinction learning may be detrimental during treatments of anxiety disorders if therapy-induced fear reductions do not generalize beyond the therapeutic setting (*Maren et al., 2013*).

Much of our fundamental understanding of the formation of fear memory traces and their suppression during extinction learning has been derived from optogenetic studies in rodents, which describe the formation and modification of fear engrams with valence and context representations in the amygdala and hippocampus, respectively (*Liu et al., 2015*; *Josselyn et al., 2015*; *Redondo et al., 2014*). These studies further show that extinction learning depends on plasticity of hippocampal context representations (*Redondo et al., 2014*) as well as on prefrontal cortex engrams (*Ramanathan et al., 2018*; *Gu et al., 2022*; *Lissek and Tegenthoff, 2024*).

In humans, neuroimaging studies have reported activation of a canonical 'fear network' during acquisition (with prominent roles of the dorsal anterior cingulate cortex and insula, and a more inconsistent role of the amygdala; *Fullana et al., 2016*), and recruitment of the hippocampus and ventromedial prefrontal cortex during extinction (*Fullana et al., 2016*; *Maren et al., 2013*). Taken together, these findings putatively reflect context dependency and safety learning, respectively (*Maren et al., 2013*). Indeed, meta-analyses have shown that despite some moderate overlap, the brain regions involved in extinction learning differ substantially from those involved in fear acquisition (*Maren et al., 2013*). Moreover, reversal – involving a change in contingencies rather than the mere absence of a US – particularly engages regions involved in prediction error detection and cognitive flexibility, such as the dorsomedial and lateral prefrontal cortex (*Wisniewski et al., 2023*; *Xin et al., 2024*). This points towards the inhibition of previously threatening stimuli via executive control during reversal, a process not typically observed in standard extinction paradigms.

While mass-univariate functional magnetic resonance imaging (fMRI) activation studies have been instrumental in identifying the brain regions involved in fear learning and extinction, they are insensitive to the patterns of neural activity that underlie the stimulus-specific representations of threat cues and contexts. Contrastingly, multivariate pattern analysis methods, such as representational similarity analysis (RSA; *Kriegeskorte et al., 2008*), have emerged as a powerful tool to investigate the content and structure of these representations (e.g. *Hennings et al., 2022*). This approach allows us to characterize the 'representational geometry' of a set of items – i.e., the structure of similarities and dissimilarities between their associated neural activity patterns. This geometry reveals how the brain organizes information, for instance, by clustering items that are conceptually similar while separating those that are distinct. This method has provided important novel insights into the representational signatures that support the formation, stabilization, and possible subsequent refinement and modification of memory traces (*Rissman and Wagner, 2012*; *Heinen et al., 2024*). This has informed our understanding of the basic mechanisms of learning and memory, while also contributing to more mechanistic theories of memory distortions in mental disorders. For example, *Visser et al., 2011*; *Visser et al., 2013* demonstrated that trial-by-trial similarities of blood oxygen level-dependent (BOLD) patterns increase during associative learning in regions of the fear network, such as the anterior cingulate cortex (ACC), ventromedial prefrontal cortex (vmPFC), or superior frontal cortex. Similar representational signatures of 'cue generalization' – i.e., increasing levels of similarity among the

memory traces of different items associated with the same valence – were observed in the amygdala related to memories of a stressful episode (*Bierbrauer et al., 2021*), as well as in sensory regions and areas of the salience network for aversive trauma-analogue stimuli (*Kobelt et al., 2024*). Furthermore, RSA can be used to study how specific neural patterns are reactivated during memory, a mechanism also referred to as 'encoding-retrieval similarity' (e.g. *Kobelt et al., 2024*). For example, *Hennings et al., 2022* observed a selective reactivation of fear versus extinction memories in the medial PFC and hippocampus depending on encoding context. Furthermore, we can measure the consistency of a neural pattern for a given item across multiple presentations. This metric, which we refer to as 'item stability,' quantifies how consistently a specific stimulus (e.g. the image of a kettle) is represented in the brain across multiple repetitions of the same item. Notably, higher item stability has been linked to successful episodic memory encoding (*Xue, 2018*; *Xue et al., 2010*). Finally, the difference between item stability and item generalization, commonly referred to as 'specificity' (*Xue et al., 2010*; *Xue et al., 2013*; *Zheng et al., 2018*; *Sommer et al., 2022*), quantifies the amount of item-specific information in a representation. This representational property could be particularly fruitful as a means to study the influence of fear reversal or extinction on context representations, which, despite some notable exceptions (e.g. *Hennings et al., 2020*), have been less systematically investigated than cue representations across different learning stages.

Here, we aimed to systematically investigate how the neural representations of cues and contexts change across different phases of learning. The phases include acquisition, reversal, and two consecutive test phases with new contexts and previous contexts, respectively, in which all cues are extinguished. We presented the CS cues in each phase in multiple different contexts that changed between phases, which allowed us to study the role of context specificity by comparing the similarity between same vs. different contexts in each phase (see *Figure 1*). For clarity, we use the term 'test' interchangeably with 'test phase' throughout the manuscript.

We hypothesized that the representational geometry of CS cues changes across learning phases, reflecting the inhibition of fear memories during reversal, as well as the formation of novel memories of cues with updated contingencies. More specifically, we expected cue generalization effects in regions of the fear network, item stability in areas related to episodic memory, and context-specific representations in the hippocampus and PFC. Finally, we hypothesized that increased context specificity during reversal would influence the reinstatement of fear memory traces during the test phases.

To test our hypotheses regarding the representational geometry of threat and safety, we used a multi-day fMRI paradigm that dissociates cue-specific learning from contextual modulation. The task was presented through a narrative ('Nina the Unlucky Backpacker') to provide an ecologically valid framework for fear acquisition and reversal.

We employed a trace conditioning design, using a temporal gap between stimulus and reinforcement to specifically engage hippocampus-dependent memory systems. In each trial, participants viewed a Context (2 s video of a natural scene) followed by a CS (1 s image of a household appliance) embedded within that scene. Throughout the task, participants provided real-time US expectancy ratings, allowing us to correlate neural representational changes with behavioral indices of learning.

The experiment was divided into four phases across two days to capture the evolution of memory traces (see *Figure 1* for a detailed schema). Day 1 (Acquisition and Reversal): We first established fear associations (Acquisition) and subsequently altered them (Reversal). This created four distinct functional cue types: stable threat (CS++), extinguished threat (CS+-), newly acquired threat (CS-+), and stable safety (CS--). Day 2 (Test phases). To assess the context-dependency of these memories, we conducted two test phases under extinction (no US delivery): In the 'Test$_{new}$' phase, CS cues were presented in novel natural scenes; in the 'Test$_{old}$' phase, cues were returned to their original acquisition or reversal contexts, while the US remained absent.

By presenting each CS across multiple different contexts and repetitions within each phase, we were able to apply RSA methods to quantify item stability (consistency of a cue's representation across repetition), cue generalization (similarity between cues of the same valence), and context specificity (the difference between context stability and context generalization, quantifying the amount of context-specific information for each phase). The following Results section details how these representational metrics evolved as Nina (and the participants) learned to navigate changing threats across different environments. For a full description of the stimulus sets, counterbalancing procedures, and the statistical procedures, please refer to the Methods section at the end of the manuscript.

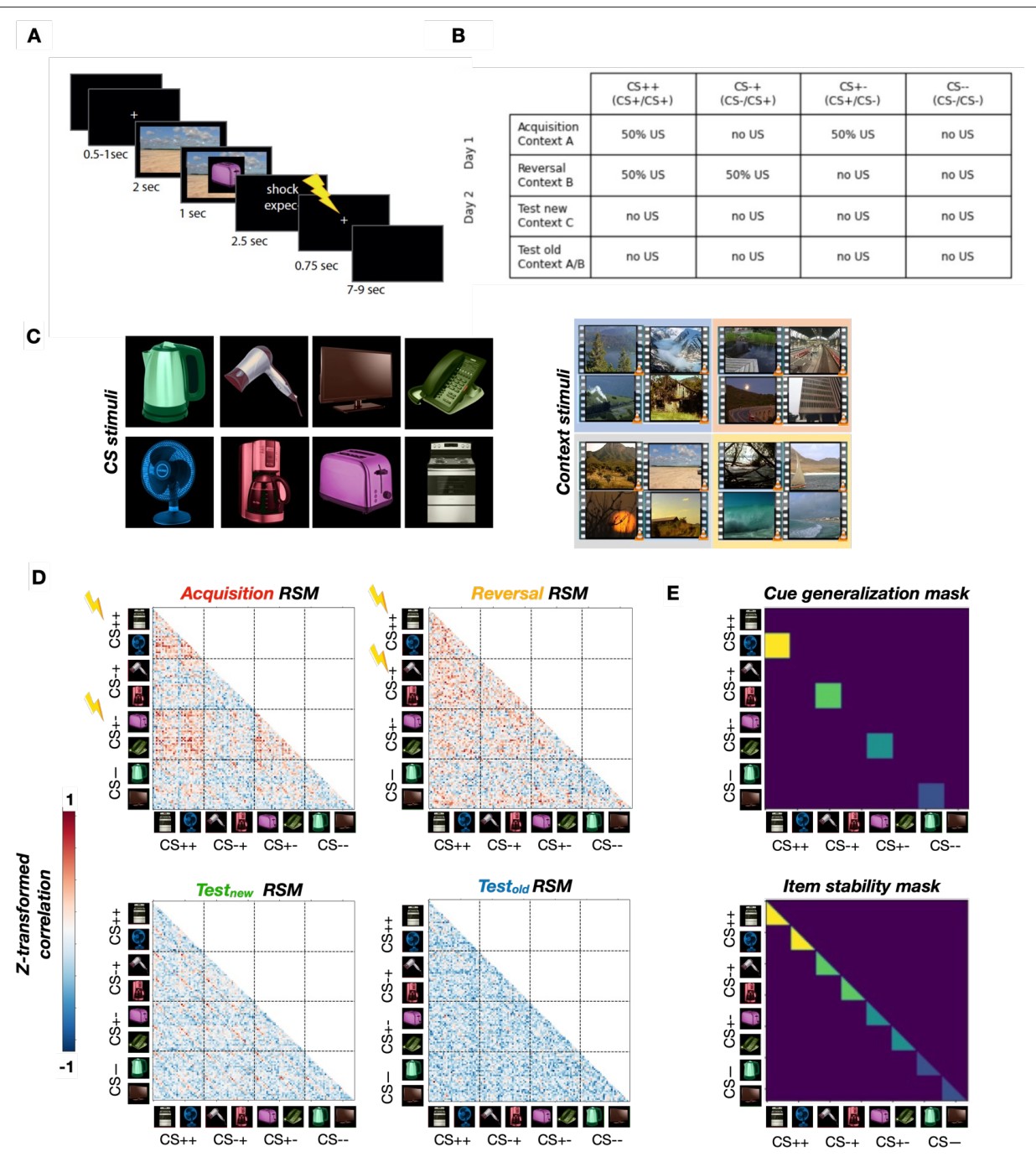

**Figure 1.** Overview of the paradigm and analysis approach. (A) Example structure of a trial. Each trial comprises the presentation of a context video, cue, and unconditioned stimulus (US) expectancy rating. Electric shocks (US) are administered in reinforced trials during acquisition (following CS++ and CS+- cues) and reversal (following CS++ and CS-+ cues), with reinforcement rates of 50%. (B) Paradigm structure with four different experimental phases (rows) and four different cue types (columns). Each cue type consists of two possible items. (C) Conditioned stimulus (CS) items (left) and context videos (right). Each color indicates a set of four thematically related context videos. Different sets are used across phases (see Table in B). (D) Representational Similarity Matrices (RSMs) for each experimental phase, shown here from the dorsal anterior cingulate cortex (ACC) for illustrative purposes. Lightning images represent reinforced cue types in the different learning phases. Representations of threatening cues are more similar to each other (warmer colors), reflecting cue generalization. (E) Top: Cue generalization mask for the representational similarity analysis (RSA) matrices estimated within each searchlight. The mask is superimposed on the RSMs (shown in C) to compute the average similarity between the different cues of each CS type (different colors). Average cue generalization values are then compared between CS types. Bottom: Item stability mask estimated within each searchlight. The mask is superimposed on the RSMs to compute the average similarity across trials of each cue, separately for each CS type (different colors). Average item stability values are then compared between CS types.

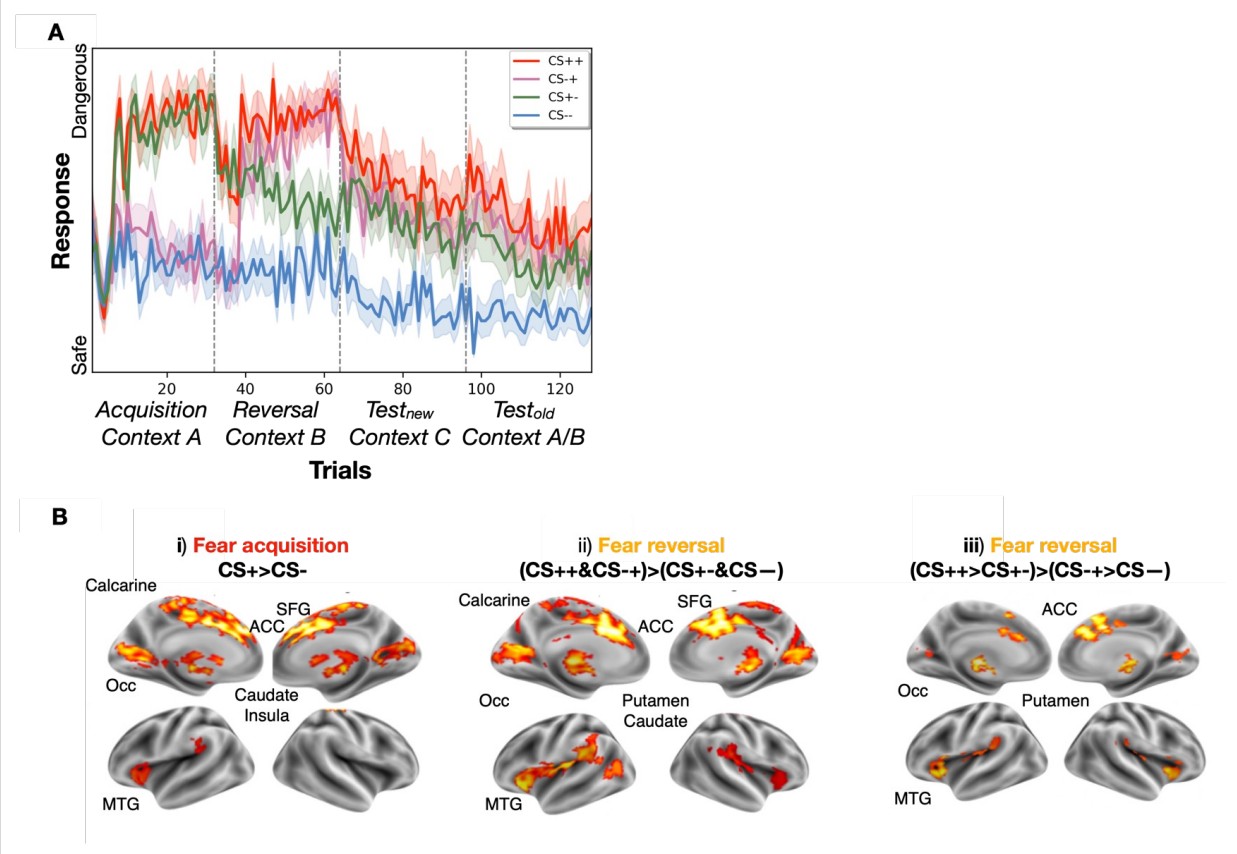

**Figure 2.** US expectancy ratings and univariate activation results. (**A**) Unconditioned stimulus (US) expectancy ratings and univariate activity difference between cue types across experimental phases. Dotted lines separate the four experimental phases. Participants quickly learned the contingencies of each cue type and their changes across the experimental phases. (**B**) Univariate activation results. Significant second-level results are shown for different contrasts in the different experimental phases. Significance was assessed at the cluster level with 10 k permutations ($p_{uncorr}$ <0.001).

## Results

### Behavioral results

We first examined the trial-wise US expectancy ratings across experimental phases. A linear mixed effects (LME) model with 'CS type' and 'experimental phase' as fixed effects and 'participant' as a random effect revealed significant effects of CS type ($F_{(1816.6, 605.54)}=479.35$, $p<0.0001$) and experimental phase ($F_{(476.3, 158.78)}=125.6$, $p<0.001$) as well as a significant interaction ($F_{(334.8, 37.2)}=29.45$, $p<0.001$), showing that both CS type and experimental phases affected US expectancy (*Figure 2A*). Post-hoc paired Wilcoxon tests (Bonferroni-corrected) showed that ratings to all CS types were significantly different from each other across all the experimental phases (CS++ > CS+- > CS-+ >CS--; all $p<0.01$) except during fear acquisition, in which CS ++ and CS+- cues on the one hand, and CS-+ and CS-- cues on the other hand were equivalent, as expected at this stage. Indeed, during fear acquisition, CS+- and CS ++ on the one hand, and CS-+ and CS-- on the other, are still functionally identical, since fear acquisition occurs before the valence change of CS-+ (newly acquired threat: from CS- to CS+) and CS+- (extinguished threat: from CS + to CS-) operated during fear reversal. Post-hoc pairwise comparisons between experimental phases across all CS types were significant as well (reversal >acquisition > test_new >test_old; all $p<0.0001$), and fear reversal and acquisition were the experimental phases with the highest US expectancy. The CS type with the highest US expectancy was, as expected, CS++.

To examine the interaction between US expectancy and CS type, we conducted post-hoc Wilcoxon tests (Bonferroni-corrected), which revealed different patterns of CS differences between experimental phases (*Supplementary file 1a*).

### Activation of fear network by cues signaling current and prior threats

Next, we assessed activity differences between CS types during each experimental phase (*Figure 2B*). During acquisition, the CS+ > CS- contrast (aggregating CS ++ and CS+- on the one hand, CS-+ and CS-- on the other) showed significantly increased BOLD activity in several clusters across the fear network, such as the dACC, superior frontal gyrus, caudate nucleus, and middle temporal gyrus (MTG; see *Figure 2Bi*), in line with previous work (e.g. *Fullana et al., 2016*). The opposite contrast CS- > CS+ showed no significant clusters.

During fear reversal, a contrast of current valence, i.e., (CS ++ and CS-+) > (CS+- and CS--) showed activation patterns similar to those during fear acquisition, spanning across the fear network (*Figure 2Bii*), reflecting the newly learned status of 'newly acquired threat' for CS-+ and of 'extinguished threat' for CS+-. We then contrasted currently threatening and safe cues depending on their previous valence (i.e. their fear acquisition valence), (CS++ > CS+-) > (CS-+ > CS--), which also revealed activation in the fear learning network, although to a lesser extent (*Figure 2Biii*). This result may reflect the impact of the lingering fear memory trace (remaining from acquisition) and/or the time required to learn contingency changes during reversal.

During the two test phases, none of the contrasts between CS types (CS++ > CS--, CS-+ > CS--, CS+- > CS--) revealed any significant activity differences. Thus, BOLD responses were similar for all CS types in the absence of a US, even though differences in US expectancy ratings were observed at the behavioral level. This further underlines the necessity for an analysis of representational patterns rather than mere univariate activity differences.

### Generalized representations of threat cues during acquisition and reversal

We thus focused our analyses on the representational geometry of cues across experimental phases. We examined the effect of cue type on two distinct representational properties, cue generalization (between-cue similarity) and item stability (within-cue similarity), using a whole-brain searchlight approach (*Figure 3A*).

During fear acquisition, we again combined CS ++ and CS+- cues (both followed by a US in 50% of trials) into a common CS + category, and CS-- and CS-+ cues (never followed by a US) into a common CS- category. We found that item stability did not differ between CS + and CS- cues. Importantly, however, cue generalization was significantly higher for CS + compared to CS- cues in several clusters across the fear network, with a pattern reminiscent of the results during the corresponding univariate analyses. Thus, the dACC, superior frontal gyrus, caudate nucleus, and insula were among the regions showing higher cue generalization of CS + compared to CS- cues (*Figure 3C*; *Supplementary file 1b*). This suggests the formation of a higher-order association (i.e. a category-level stability) between threatening cues and less so between safe cues during fear acquisition. The opposite contrast (CS- > CS+) did not reveal any significant effects.

### Distinct functional roles, spatial distributions, and subsequent persistence of item stability and cue generalization during reversal

Next, we compared item stability and cue generalization between the four CS types during fear reversal (*Figure 3B–D*). We first compared CS++ and CS-- cues, corresponding to the CS+ vs. CS- contrast during fear acquisition (*Figure 3B*). Again, we observed higher cue generalization for CS++ compared to CS-- cues in the dACC, but not in any of the other regions where cue generalization effects were observed during acquisition. Comparing the cue generalization of all CS cues that are threatening in reversal (CS++ and CS-+) to the ones that are not (CS+- and CS--) revealed similar results to fear acquisition, with increased cue generalization across fear learning network regions, including the dACC, superior frontal gyrus (SFG), medial temporal gyrus, and inferior frontal gyrus (IFG) (*Figure 3C*). Item stability, in line with results from fear acquisition, did not differ between currently threatening and non-threatening cues. Moreover, we found no significant clusters when comparing item stability between CS++ vs. CS-- cues.

We then investigated representational effects of contingency changes and compared cues that changed their contingency between acquisition and reversal (CS+- and CS-+) to cues with consistent contingencies (CS-- and CS++), resulting in the 'change vs. no-change' contrast, i.e., (CS+- & CS-+) > (CS++ and CS--). Item stability was significantly higher for changing than consistent cues in the

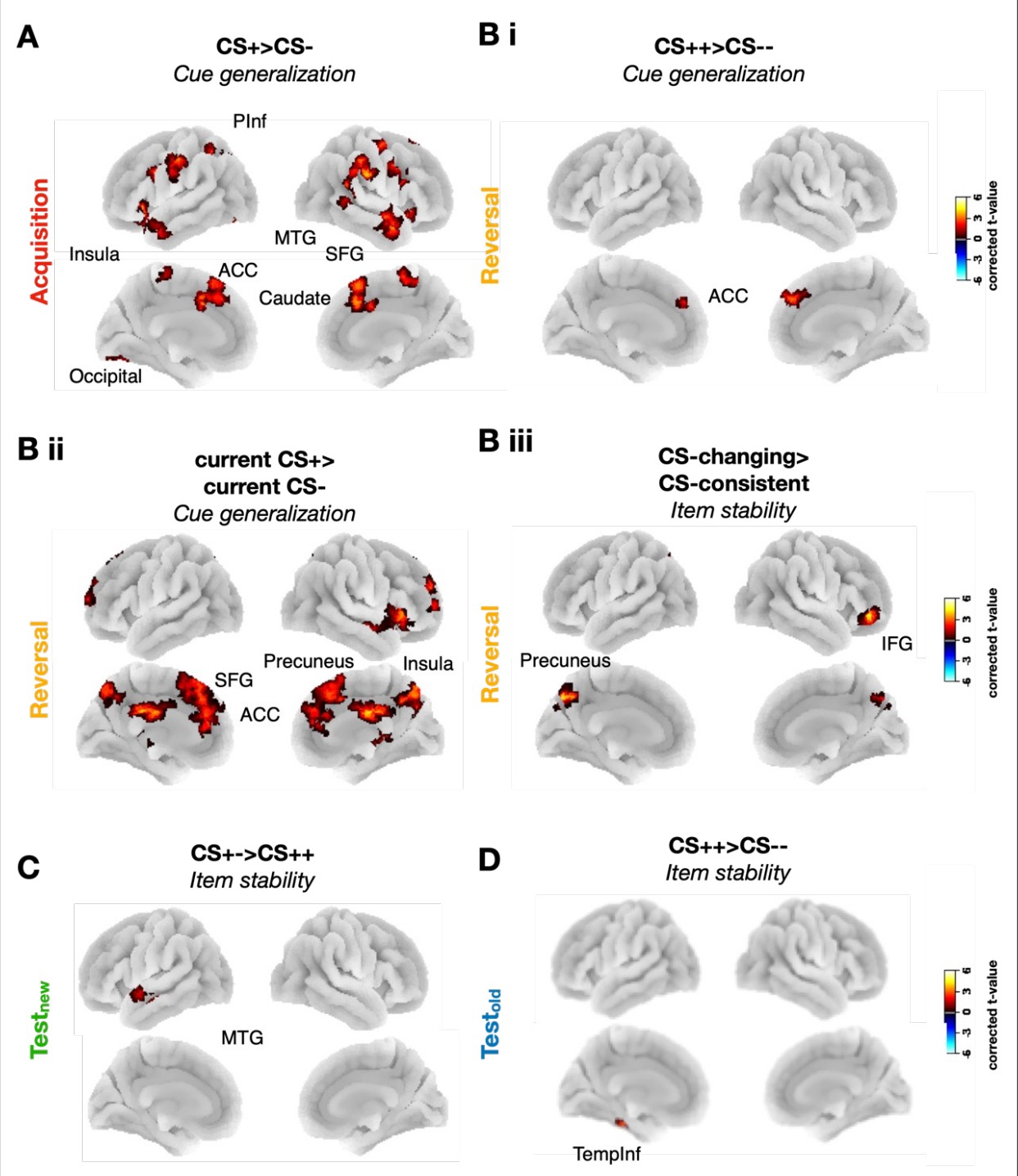

**Figure 3.** Enhanced cue generalization and item stability of threat cues. (**A**) Cue representations during acquisition showing higher cue generalization of conditioned stimulus (CS)+ than CS- cues. No differences in item stability were found. (**B**) Cue representations during reversal. (**Bi**) Higher cue generalization of CS++ than CS-- cues. (**Bii**) Higher cue generalization of currently threatening than non-threatening cues i.e., (CS-+ and CS++) > (CS+- and CS--). (**Biii**) Higher item stability of cues with changing valence than cues with consistent valence, i.e., (CS-+ and CS+-) > (CS++ and CS--). (**C**) Cue representations during test_new showing higher item stability of previously safe cues vs. always threatening cues (CS+-) > (CS++). (**D**) Cue representations during test_old showing higher item stability of 'always threatening' vs. 'never threatening' cues (CS++) > (CS--). All plots depict t-values from searchlight analyses within family-wise error-corrected clusters (uncorrected $p<0.001$, corrected $p<0.025$ for Acquisition, and $p<0.0125$ for the other phases) with 10 k permutations.

precuneus and IFG, i.e., in regions that overlapped with those showing higher cue generalization for threatening cues (*Supplementary file 1c*; *Figure 3Biii*). Conversely, this contrast did not reveal any differences in cue generalization.

We next investigated item stability and cue generalization during test$_{new}$ and test$_{old}$, i.e., when USs are absent for all CS types. In order to understand the impact of prior contingencies – i.e., of lingering fear memory traces – on test$_{new}$ and test$_{old}$, we examined the contrast between CS++, CS-+, and CS+- with CS-- (safe baseline), as well as the contrast between CS-+ and CS+- with CS++ (unsafe baseline) in these two phases.

Cue generalization did not differ between CS types during either test$_{new}$ or test$_{old}$. However, item stability was higher for CS+- vs. CS ++cues in a middle temporal cluster during test$_{new}$ (*Figure 3E*), and for CS++ compared to CS-- cues in an inferior temporal cluster during test$_{old}$ (*Figure 3F*; *Supplementary file 1d*).

To summarize, cue generalization and item stability showed a dissociation during reversal, with higher cue generalization for threatening vs. non-threatening cues and higher item stability for changing vs. consistent cues. This suggests that these two representational properties could capture distinct aspects of contingency learning; the threatening (vs. safe) nature of cues increased cue generalization, while a changing (vs. consistent) nature of contingencies enhanced item stability. We also found that item stability, but not cue generalization effects, persisted in the absence of a US.

## Memory traces from previous learning phases compete for reinstatement during test

Fear extinction is commonly described as an inhibitory process, in which a new safety memory trace is created and competes with the previous threat memory trace that is concurrently inhibited (*Lebois et al., 2019*; *Santini et al., 2008*; *Szeska et al., 2020*). In our paradigm, the memory traces formed during acquisition and reversal might compete for reinstatement, particularly during the test$_{old}$ phase, because both fear and reversal memories may reoccur during this phase due to context overlap. We compared the magnitude of reinstatement effects between acquisition and reversal during test$_{old}$, by using an LME with experimental phase, CS type, and their interaction as predictors, and reinstatement as the predicted variable. Reinstatement was estimated by comparing either the similarities of identical items across phases (item reinstatement) or the similarities of different items from one cue type across phases (generalized reinstatement). We extracted these reinstatement values from significant clusters observed in our previous searchlight analyses (*Figure 4A*) and correlated them across participants (see *Graner et al., 2020* for a similar approach).

We observed a significant effect of experimental phases on item reinstatement in IFG (F(2,253)=5.50, $p<0.01$) (*Figure 4Bi*). Post-hoc Wilcoxon t-tests showed that reversal-test$_{old}$ item reinstatement was significantly higher than acquisition-test$_{old}$ item reinstatement (t(253)=-3.01, $p<0.01$) and test$_{new}$-test$_{old}$ item reinstatement (t(253)=-2.7, $p<0.05$). In addition, we observed a significant effect of experimental phases on generalized reinstatement in dmPFC (F(2,259)=4.01, $p<0.05$) (*Figure 4Bii*), where acquisition-test$_{old}$ generalized reinstatement was higher than test$_{new}$-test$_{old}$ generalized reinstatement (t(259)=2.96, $p<0.05$). In summary, during the test$_{old}$, we observed prominent reinstatement of item-specific reversal memory traces in IFG and of generalized acquisition memory traces in dmPFC, suggesting that memories from these two phases tend to come back in different representational formats and in dissociable brain regions.

## Context specificity increases during reversal in prefrontal cortex and predicts the reoccurrence of fear memory traces

We followed our analyses of cue generalization, item stability, and reinstatement with exploratory analyses of the neural representation of contexts across experimental phases. Given that memories built during extinction learning tend to be more context-specific than those acquired during initial fear learning (*Maren et al., 2013*), we established a measure of context specificity, namely the difference between neural representations of same vs. different contexts in each phase. We also compared this measure across experimental phases and related it to the representational geometries of cues (*Figure 5A*).

As hypothesized, we found that context specificity during reversal was significantly higher than it was during acquisition, an effect that occurred in a cluster, including both dorsomedial PFC and

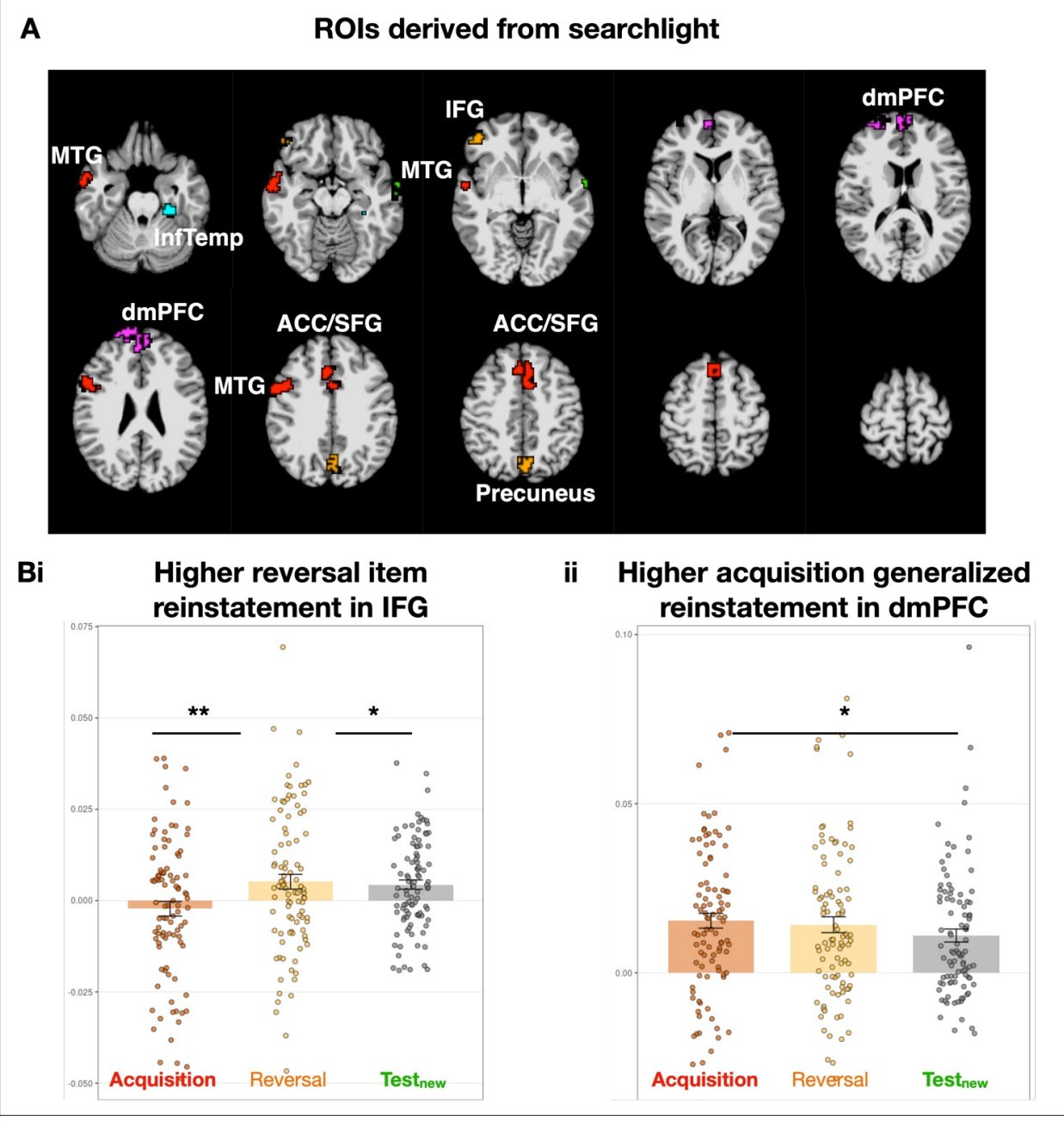

**Figure 4.** Different reinstatement patterns are observed for the previous experimental phases during Test$_{old}$. (**A**) Region of interests (ROIs) derived from the previous searchlight analyses (see **Figure 3**), by extracting the significant clusters from the previous statistical analyses. ROIs are color-coded depending on the experimental phase they are derived from: red for acquisition, orange for reversal, green for test$_{new}$, blue for test$_{old}$. When several ROIs overlapped, only the ROI with the bigger voxel size was included in the analyses. MTG: Middle Temporal Gyrus. InfTemp: Inferior Temporal Gyrus. IFG: Inferior Frontal Gyrus. dmPFC: Dorsomedial Prefrontal Cortex. ACC: Anterior Cingulate Cortex. SFG: Superior Frontal Gyrus. (**B**) Reinstatement during test$_{old}$ differed between experimental phases, such that: (**Bi**) in IFG, item reinstatement was higher for memory traces from reversal compared to those from acquisition and test$_{new}$; and (**Bii**) in dmPFC, generalized reinstatement was higher for memory traces from acquisition compared to those from test$_{new}$. *:$p < 0.05$. **:$p < 0.01$. n=255 for Bi and n=261 for Bii. Errors bars represent standard error.

lateral PFC (i.e. superior frontal gyrus), areas known to be involved in contextual processing (**Maren et al., 2013**; **Figure 5B**, **Supplementary file 1e**). Context specificity did not differ between the other experimental phases.

Previous research has suggested that higher context specificity during extinction learning predicts a more pronounced reoccurrence of fear memories (**LaBar and Phelps, 2005**; **Milad et al., 2005**;

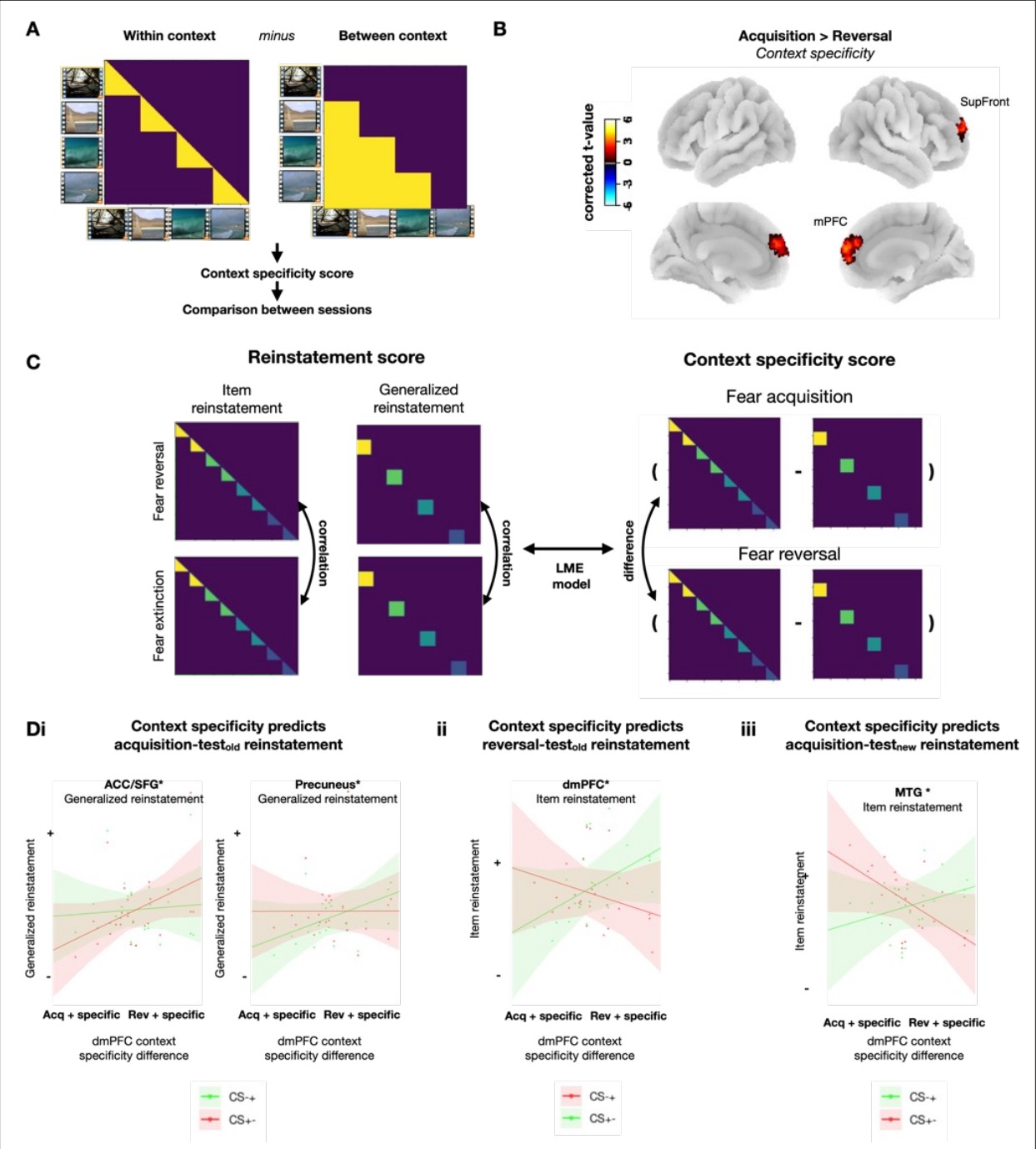

**Figure 5.** Context specificity during reversal and its role for reinstatement of fear memory traces. (**A**) Calculation of context specificity as the difference of within-context similarity and between-context similarity. (**B**) Difference in context specificity between acquisition and reversal. Positive values indicate higher context specificity in reversal. (**C**) Calculation of item reinstatement and generalized reinstatement (similarities of item representations across different phases; left) and context specificity (difference between acquisition and reversal; right). An LME model was used to predict these reinstatement measures by the interaction of context specificity and conditioned stimulus (CS) types. (**D**) Higher context specificity during reversal predicted reinstatement during test$_{old}$, as a function of CS type: (**Di**) Higher reversal context specificity predicted more pronounced generalized reinstatement of CS+- vs. CS-+ acquisition memory traces in anterior cingulate cortex (ACC)/superior frontal gyrus (SFG) (left), and reversely, more pronounced generalized reinstatement of CS-+ vs. CS+- acquisition memory traces in the precuneus (right). CS+-, which is threatening in acquisition, is shown in red, and CS-+, which is not threatening in acquisition, is shown in green. (**Dii**) Higher reversal context specificity predicted more pronounced item reinstatement of CS-+ than CS+- reversal memory traces in dorsomedial Prefrontal Cortex (dmPFC). CS-+, which is threatening in acquisition, is shown in red, and CS+-, which is not threatening in acquisition, is shown in green. (**Diii**) Higher reversal context specificity also predicted more pronounced item reinstatement of CS-+ than CS+- memory traces from reversal in MFG during test$_{new}$.

*Vansteenwegen et al., 2005*; *Neumann, 2006*; *Navarro-Sánchez et al., 2024*). To investigate this hypothesis, we compared the neural representations of cues between experimental phases and correlated the magnitude of reinstatement effects with the increase in context specificity from acquisition to reversal (across participants). We extracted subject-wise measures of context specificity from the PFC cluster in *Figure 4B* and compared them to both item reinstatement and generalized reinstatement. We extracted these reinstatement values from significant clusters resulting from our previous searchlight analyses (*Figures 4A and 5C*).

We focused our analyses on cues that changed their contingencies (i.e. CS-+ and CS+-), which were expected to reveal differential reinstatement of acquisition vs. reversal memory traces. We used LME models with 'context specificity' and 'cue type' as predictors and 'item reinstatement' or 'generalized reinstatement' as the dependent variable (*Figure 5C*). Correction for multiple comparisons was done (FDR) within each item reinstatement/generalized reinstatement pair of each ROI.

We tested whether increased context specificity during reversal predicted the reinstatement of acquisition memory traces during test$_{old}$ (the test phase with the acquisition/reversal contexts). We found that an interaction between context specificity and cue type predicted generalized reinstatement of acquisition memory traces in both ACC/SFG (t(22)=6.25, *p*<0.05) and precuneus (t(22)=4.89, *p*<0.01) (*Figure 5Di*). In the precuneus, higher context specificity during reversal predicted more generalized reinstatement of the initially non-threatening cues (CS-+), i.e., a cue type that is safe during both acquisition and test, as compared to initially threatening cues (CS-+). Reversely, in the ACC/SFG, higher context specificity during reversal predicted more generalized reinstatement of the initially threatening cues (CS+-) than of the initially safe cues (CS-+).

In addition, higher context specificity during reversal predicted higher item reinstatement of reversal memory traces for CS-+ than CS+- cues in the dmPFC (t(22)=5.56, *p*<0.05). Thus, similar to reinstatement in ACC/SFG, higher context specificity again favored reinstatement of memory traces of threatening over safe cues, even though these memory traces were now from the reversal rather than the acquisition phase (*Figure 5Dii*).

As a control, we also analyzed reinstatement during the test$_{new}$ phase. We tested whether increased context specificity during reversal predicted the reinstatement of acquisition memory traces during this phase with new contexts. We found that an interaction between context specificity and CS type predicted the reinstatement of acquisition memory traces in MTG (t(22)=2.51, *p*<0.05) (*Figure 5Diii*). Specifically, higher reversal context specificity predicted more item reinstatement of CS-+ cues, i.e., cues that were safe during acquisition, than of the initially threatening CS+- cues.

Together, these results indicate more specific context representations during reversal than acquisition (*Figure 5B*) and show that acquisition memory traces are predominantly reinstatement in a generalized format, while reversal memory traces are reinstated at the level of individual items (*Figure 4B*). They suggest a possible mechanism for the previously observed impact of extinction contexts on fear renewal, because higher levels of context specificity during reversal favored the reinstatement of threat memory traces in areas of the fear network (ACC and dmPFC; *Figure 5Di* left and *Figure 5Dii*). These effects were not observed in the precuneus (*Figure 5Di* right) and for reinstatement during new contexts (*Figure 5Diii*).

## Discussion

The present study investigated the dynamic changes in neural representations of cues and contexts during acquisition, reversal, test in new contexts (test$_{new}$), and test in previous acquisition/reversal contexts (test$_{old}$). Our main findings demonstrate distinct representational properties of CS cues and contexts during these different phases, suggesting that representational geometries reflect the fate of memory traces. We found that (1) cue generalization and item stability play complementary roles during initial fear learning and reversal, by being associated with threatening-vs-safe cues and changing-vs-consistent cues, respectively; (2) during test$_{new}$ and test$_{old}$, differences of cue generalization between CS types disappear, while some differences of item stability remain; (3) context representations become more specific following contingency changes during reversal learning, and (4) the context specificity during reversal predicts the reinstatement of fear memories during subsequent tests, providing a mechanistic basis for clinically relevant phenomena, such as renewal. These results offer new insights into the regional distributions, representational geometries, and functional

relevance of cues and contexts across distinct stages of fear learning, opening new avenues of understanding fear-guided behavior.

## Complementary representational properties during initial fear learning

Our results demonstrate how different representational properties of CS cues are associated with distinct aspects of fear learning: cue generalization with the threatening vs. safe nature of the CS, and item stability with the changing vs. consistent nature of the CS. Consistent with previous studies (*Visser et al., 2011*; *Visser et al., 2013*), we found that cue generalization was greater for CS + than for CS- cues during fear acquisition in regions of the fear network (e.g. ACC) and salience network. This suggests that fear acquisition leads to the formation of a higher-order association between different reinforced cues, but less so between unreinforced ones. This category-level learning could allow for efficient threat detection and generalization — an adaptive behavior in potentially dangerous environments. Moreover, previous studies showed that the role of cue generalization in the coding of threat extends beyond fear conditioning, as shown by *Dunsmoor et al., 2015* who found enhanced memory consolidation of items sharing conceptual similarity with threat-associated stimuli.

In stark contrast, item stability of CS cues, i.e., their within-stimulus similarity across repetitions, did not differ between CS + and CS- cues during fear acquisition. This indicates that while acquisition induces a category-level representation for reinforced cues, it does not differentially modify the item-level representations of CS + compared to CS- cues. Contrastingly, item stability was particularly sensitive to changes in CS valence between experimental phases, suggesting that it plays a crucial role in tracking and updating the specific threat associations of individual stimuli. Indeed, in fear reversal, cue generalization remained greater for reinforced compared to unreinforced cues (CS--), mirroring the pattern observed during acquisition. However, item stability specifically increased for cues that changed valence between acquisition and reversal (CS-+ and CS+-). This finding suggests that when the contingencies change, the participants might focus more on the individual properties of cues to interpret the new contingencies, leading them to fine-tune their representations. Indeed, item stability has been linked to successful memory encoding and retrieval in several studies (*Xue et al., 2010*; *Zheng et al., 2018*). Neural correlates of item stability have been reported in regions of the episodic memory network, such as the IFG and precuneus (*Xue et al., 2010*), where we found a significant effect of item stability during reversal. Therefore, item stability might be more akin to an episodic-like type of learning, while cue generalization might be more reflective of category-level learning (*Visser et al., 2013*). Furthermore, our use of a trace conditioning paradigm, which is known to engage the hippocampus more than delay conditioning does, may have facilitated the detection of item-specific, episodic-like memory traces and their interaction with context. This strengthens the relevance of our findings for understanding the interplay between aversive learning and the mechanisms of episodic memory.

Previous findings by *Visser et al., 2011*; *Visser et al., 2013* demonstrate distinct learning curves for item stability between CS + and CS- cues from trial to trial. This discrepancy could be caused by methodological differences, as our study focused on session-wise differences of item stability for each cue type and not on trial-by-trial differences. In line with our conclusions, however, *Visser et al., 2013* found that item stability was increased for subsequently remembered cues, while cue generalization was associated with the later behavioral expression of fear memory. Overall, the increased item stability during reversal of the items that change contingency could reflect a process of stabilizing the new valence at the item level. This is because the change of contingency may lead to the temporary representation of individual items as 'categories' themselves, without being formed yet into generalized representations encompassing multiple different items sharing the same valence. This dual representational signature may allow for both efficient threat detection (via category representations) and flexible updating of individual stimulus associations (via item-specific representations).

## Dissolution of cue generalization and item stability in the absence of US

During the test phases, we did not observe any differences in cue generalization between cue types. However, some differences in item stability remained during $test_{new}$ (higher for CS+- vs. CS ++in the MTG) and $test_{old}$ (higher for CS ++ vs. CS-- in the inferior temporal cortex) (*Figure 3C–D*).

These findings suggest that the disappearance of threat during the test phases may involve two concurrent processes: (1) An unlearning of generalized threat representations, evidenced by the absence of cue generalization differences during the test phases; and (2) a partial unlearning of item-level representations, particularly for cues with changing contingencies, reflected in diminished item stability. Interestingly, these effects occurred during the test phases rather than during reversal, suggesting that they are driven by the absence of the US rather than by the contingency change. During the reversal, the continued presence of the US, albeit with a different contingency, may still benefit from generalized representations at the item and category levels. Contrastingly, the complete absence of the US during the test phases may promote a differentiation of CS representations, as the need for generalization diminishes. This finding highlights the importance of the specific reinforcement history of cues on the dynamics of fear representations.

## Reinstatement of item representations during the test is weaker for fear extinction

Our results showed a differentiation of fear memories during the test phases, both at the item and category level. Interestingly, representations from the first test phase were less reinstated during the second test phase (despite the fact that both test phases occurred during the second experimental day), compared with acquisition and reversal traces formed on the first experimental day (*Figure 4E*). This may be explained by the greater differentiation of cue representations during test$_{new}$; the more differentiated the representations, the less likely they are to be subsequently reinstated. The weaker reinstatement of memories from a phase without any US, compared to memories from acquisition and reversal phases with US, may contribute to the challenges of preventing relapse in anxiety disorders (*Vervliet et al., 2013*). If extinction learning results in less stable and less generalizable safety representations, individuals may remain vulnerable to the return of fear once they return to previous contexts (*Boschen et al., 2009*).

## Increased specificity of context representations following contingency changes

Our analysis of context representations revealed an increased specificity of context encoding during reversal compared to initial acquisition. This suggests that the brain may allocate more resources to the representation of contextual details when contingencies are changing, by potentially facilitating the adaptive updating of contingencies against a more stable contextual backdrop.

The dorsomedial PFC, including the superior frontal gyrus and ACC, have emerged from our analyses as key regions exhibiting higher context specificity in reversal learning. Given their roles in attentional control (*Dosenbach et al., 2007*) and conflict monitoring (*Stevens et al., 2011*), the dmPFC's involvement may reflect increased attentional and control demands induced by changing contingencies. Computationally, the more precise contextual encoding in these regions during reversal could serve to disambiguate cues of changing contingencies, supporting the formation of new context-dependent associations (*Xu and Südhof, 2013*). Our findings extend prior work on the importance of the hippocampus and mPFC in representing context during fear learning and extinction (*Maren et al., 2013*), as these regions could dynamically adjust their representational specificity in response to a change in environmental demands.

## Context specificity is associated with reinstatement of fear memory traces

The amount of reinstatement during the test$_{old}$ was related to the increase in context specificity from acquisition to reversal. We quantified this increase in specificity in the dmPFC cluster identified in the previous analysis and correlated it with two measures of reinstatement: (1) item reinstatement, reflecting the similarity of individual cue representations between phases; and (2) generalized reinstatement, capturing the similarity of cue representations among their CS categories.

For regions involved in threat processing, such as the ACC/SFG, higher context specificity predicted stronger generalized reinstatement of representations of previously threatening cues (CS+-) from acquisition to test. This suggests that for these cues, the more distinct the contextual encoding during reversal, the more strongly the original fear memory trace resurfaced, likely reflecting a return of fear (*Figure 5D*). Contrastingly, for areas implicated in cue-specific processing that could reflect more

episodic-like learning, such as the precuneus (*Cavanna and Trimble, 2006*), context specificity was associated with enhanced generalized reinstatement for cues with consistent meanings across phases (e.g. CS+- cues from reversal to test). Regarding item reinstatement, the dmPFC behaved similarly to the ACC/SFG, with stronger item reinstatement of previously threatening cues (CS-+ from reversal to test), while the MTG showed a pattern similar to the precuneus, with stronger item reinstatement for cues with consistent meanings across phases.

These findings highlight the region-, phase-, and cue-specific effects of contexts on the reinstatement of cue representations. In threat-responsive regions, context specificity may promote the resurgence of generalized threat representations, in line with notions of renewal and spontaneous recovery of fear (*Maren et al., 2013*). Conversely, in episodic learning regions, contextual encoding may support the reactivation of representations when meanings are maintained, reflecting memory stability. Together, these results suggest a critical role of context representations in modulating the balance between generalization and specificity of fear memories over time.

## Limitations and future directions

While our study provides novel insights into the changes of neural representations across the different stages of fear learning, reversal, and test, several limitations should be noted. First, our sample size was relatively small, and future studies with larger samples will be needed to replicate and extend our findings. Second, while we examined the spatial patterns of neural activity using RSA, we did not assess potential changes in the temporal dynamics of these patterns. Several studies have highlighted the importance of considering temporal information in understanding the neural mechanisms of fear learning (*Visser et al., 2013*; *Sperl et al., 2021*). Integrating spatial and temporal pattern analysis in future studies could provide a more comprehensive overview of how fear representations evolve over time. Moreover, further examining the role of context manipulation, by using more classical approaches where only one context is presented per phase, could extend and generalize our current findings. Finally, applying our approach to clinical populations could yield important insights into the neural mechanisms underlying the overgeneralization of fear and the impaired contextual regulation of fear responses in psychiatric disorders.

## Conclusion

Our study reveals the changes in neural representations of conditioned stimuli and contexts across fear learning phases. Cue generalization and item stability play complementary roles in fear acquisition, reversal, and test, by capturing the formation of threat-related categories and updating the contingency of individual stimulus representations, respectively. Phases devoid of US cues lead to a differentiation (or dissolution) of both category- and item-level representations. Context specificity in the prefrontal cortex modulates the persistence of fear memories, with region-specific reinstatement effects. These findings provide insights into the representational dynamics underlying fear learning and extinction, demonstrating the interplay between cue- and context-based representations in shaping the formation, updating, and reinstatement of fear memories. Understanding these mechanisms might help optimize interventions targeting pathological fear in anxiety disorders. Future research should extend these findings to clinical populations and investigate the identified representational properties as biomarkers for assessing the effectiveness of extinction-based therapies.

## Methods

### Participants

Thirty-two healthy participants were recruited via flyers and the online recruitment systems of the Faculty of Psychology at Ruhr University Bochum. As our paradigm consisted of a two-day design with two experimental phases per day, we observed some attrition between experimental phases. The number of participants with usable fMRI data (i.e. full data) for each phase was as follows: N=29 for the day 1 phases, and N=26 for day 1 and day 2 phases. An additional two participants were excluded from the analysis due to excessive head movement (>2.5 mm in any direction). This resulted in a final sample of 24 participants (8 males) between 18 and 32 years of age (mean: 24.69 years, standard deviation: 3.6) with complete, low-motion fMRI data for all analyses. All participants provided written informed consent before participation and were unaware of the aims of the experiment. The

procedures were performed in accordance with the tenets of the Declaration of Helsinki and were approved by the ethical review board of the Faculty of Psychology at Ruhr University Bochum.

## General procedure and stimuli

The paradigm was administered to participants in the MRI scanner and consisted of four experimental phases spanning two days. To make the experiment more engaging for participants, they were presented with the narrative of 'Nina the unlucky backpaper' and asked to play as the character Nina. During her fictitious trip, Nina would visit different places represented by videos of natural scenes that served as contexts. In each of these places, Nina would interact with different household appliances (the CS). Due to her misfortune, many of these items contain a serious defect, and their manipulation could result in a mild electric shock (the US) experienced by Nina, and by extension, the participant. The four experimental phases, therefore, correspond to different trips undertaken by Nina during her travels.

Each phase comprised 128 trials with a similar structure: presentation of a context (video showing a natural scene) for 2 s, followed by the CS (household appliance) embedded within the context for 1 s. US expectancy responses were then collected during a 2.5 s period using a 4-point Likert scale, followed by the delivery (or absence) of an electric shock (US). Finally, participants saw a fixation cross which served as an inter-stimulus interval (*Figure 1A*). During each phase, eight different CSs were presented across multiple repetitions. The experimental phases differed in the way the CSs were associated with a US as well as in the possible contexts in which the CSs were embedded. It is important to note that the temporal gap between the CS offset and potential US delivery (see *Figure 1A*) indicates that our paradigm employs a trace conditioning design. This form of learning is known to be hippocampus-dependent and has been distinguished from delay conditioning.

Visual stimuli were presented using the Presentation software package (Neurobehavioral Systems, Berkeley, CA, USA). Electrical stimulation was delivered using a constant voltage stimulator (STM200, BIOPAC Systems, Goleta, CA, USA). Electrical stimulation was delivered via two Ag/AgCl electrodes attached to the distal phalanx of the index and middle fingers of the non-dominant hand. The intensity of the electrical stimulation was calibrated individually for each participant prior to the experiment. Using a stepping procedure, the voltage was gradually increased until the participant rated the sensation as 'unpleasant but not painful'. A fixation cross was displayed with a jittered duration (7–9 s) to serve as an intertrial interval at the end of each trial. Each experimental phase contained 128 trials in total.

## Fear acquisition, reversal, and test

The first day of the experiment comprised two phases: fear acquisition and fear reversal (*Figure 1B*). During the fear acquisition phase, four CSs were associated with a US (CS+, each with 50% reinforcement rate), while the other four were never followed by a US (CS-). Every CS was associated with four different contexts (natural scenes) with equal likelihood (i.e. across the 128 trials of each phase, every combination of a given CS with a given context occurred four times).

The fear acquisition phase was immediately followed by the fear reversal phase. Here, participants were again presented with the same CS. However, half of the CSs that were associated with a US during fear acquisition were no longer associated with a US (CS+-), while the other half remained associated with a US (CS++). Similarly, half of the CS- cues became associated with a US (CS-+), while the other half remained unassociated with a US (CS--). Both CS++ and CS-+ were reinforced in 50% of the trials.

The second day of the experiment comprised two experimental test phases: test in new contexts and test in acquisition/reversal contexts (*Figure 1B*). During these two test phases, no CS was ever associated with a US. They differed in terms of the context videos in which the CS were embedded, with new contexts for the 'test$_{new}$' phase, and the previous acquisition and reversal contexts for the 'test$_{old}$' phase (see details below).

To summarize: CS++ cues were associated with a US during both fear acquisition and reversal; CS-+ cues were not associated with a US during fear acquisition but were during reversal; CS+- cues were associated with a US during fear acquisition but not during reversal; CS-- cues were not associated with a US during either fear acquisition or reversal.

## Relationship between CS types, contexts, and experimental phases

In all phases, the context videos and CS types were presented in different pseudorandom orders, such that they were orthogonal to each other. The first and last trials of each participant contained unreinforced cues. The assignment of each cue to the four possible CS types was counterbalanced across participants, as was the assignment of the type of videos used as context during each experimental phase.

Regarding context: One set of four videos was shown during fear acquisition (set 'A'); a new set of four videos was shown during fear reversal (set 'B'); a third set of eight videos was shown during test in new contexts (set 'C'); during test in previous contexts, the four videos shown during fear acquisition and the four videos shown during fear reversal were shown (sets 'A' and 'B,' in pseudorandom order).

Thus, the two test phases differed only in the type of context videos shown (new contexts during test_new and previously shown contexts test_old).

## Behavioral data analysis

Participants provided US expectancy ratings after the presentation of each CS by indicating on a 4-point Likert scale how dangerous or safe they perceived the CS to be. We examined the influence of experimental phase and CS type on US expectancy by averaging US expectancy ratings across all trials of a given CS type, separately for each experimental phase and participant. This resulted in four (averaged) US expectancy ratings per experimental phase per participant.

As the repeated measures ANOVA assumption of sphericity was not met by the data (Mauchly test; $W=0.39$, $p<0.0001$), we chose a linear mixed effects (LME) approach to predict the effect of cue type and learning phase on US expectancy, with subject as a random effect. We used Satterthwaite's approximation of degrees of freedom, as implemented in the R packages lme4 and lmerTest, using the restricted maximum likelihood method. Studies have shown that this provides a more robust estimate of degrees of freedom, and thus reduces the risk of type 1 error, compared to other methods, such as the likelihood ratio test (*Luke, 2017*).

As a preview of our behavioral data analyses, the results showed that participants quickly learned the contingencies in the initial fear acquisition phase, as well as the contingency changes introduced between the subsequent reversal and test_new/test_old phases (*Figure 1B*).

## MRI data collection

All MRI data were acquired on a Philips 3T Achieva scanner 3T MRI scanner (Philips Healthcare, Best, Netherlands). MRI data were acquired with simultaneous recording of electroencephalographic data and skin conductance recording in the scanner, which are not presented here. A reference structural T1 image was acquired on the first experimental day (TR = 817 ms, TE = 3.73 ms, 240×240×223 matrix, 1 mm isotropic resolution). All four experimental learning phases were performed in different sessions in the scanner with a BOLD echo-planar imaging sequence (TR = 2.53 s, TE = 30 ms, 96×96×46 matrix, 2.5 mm isotropic resolution). Furthermore, phase-opposite scans (with otherwise identical acquisition parameters) were acquired for each task session to correct for distortion artifacts.

## MRI preprocessing

MRI data were preprocessed with fMRI prep (*Esteban et al., 2019*; https://fmriprep.org/) as described below.

## Anatomical data preprocessing

The T1-weighted (T1w) image was corrected for intensity non-uniformity with N4BiasFieldCorrection (*Tustison et al., 2010*), distributed with ANTs 2.2.0 (*Avants et al., 2008*), and used as T1w reference throughout the workflow. The T1w reference was then skull-stripped with a *Nipype* implementation of the antsBrainExtraction.sh workflow (from ANTs), using OASIS30ANTs as the target template. Brain tissue segmentation of cerebrospinal fluid, white matter, and gray matter was performed on the brain-extracted T1w using fast (FSL 5.0.9, *Zhang et al., 2001*). Brain surfaces were reconstructed using *recon-all* (FreeSurfer 6.0.1, *Dale et al., 1999*), and the brain mask estimated previously was refined with a custom variation of the method to reconcile ANTs-derived and FreeSurfer-derived segmentations of the cortical gray matter of Mindboggle (*Klein et al., 2017*). Volume-based spatial normalization to standard space (MNI152NLin2009cAsym) was performed through nonlinear registration

with *antsRegistration* (ANTs 2.2.0), using brain-extracted versions of both the T1w reference and T1w template. The following template was selected for spatial normalization: *ICBM 152 Nonlinear Asymmetrical template version 2009c* (***Fonov et al., 2009***).

## Functional data preprocessing

For each of the four BOLD sessions per subject, the following preprocessing was performed. First, a reference volume and its skull-stripped version were generated using a custom methodology of *fMRIPrep*. Head-motion parameters with respect to the BOLD reference (transformation matrices, and six corresponding rotation and translation parameters) were estimated before any spatiotemporal filtering using mcflirt (FSL 5.0.9, ***Jenkinson et al., 2002***). BOLD sessions were slice-time corrected using 3dTshift from AFNI 20160207 (***Cox and Hyde, 1997***, RRID:SCR_005927). A deformation field to correct for susceptibility distortions was estimated based on *fMRIPrep*'s *fieldmap-less* approach. The deformation field results from co-registering the BOLD reference to the same subject's T1w reference with its intensity inverted (***Wang et al., 2017***; ***Huntenburg, 2014***). Registration was performed with antsRegistration (ANTs 2.2.0), and the process was regularized by constraining deformation to be nonzero only along the phase-encoding direction and modulated with an average fieldmap template (***Treiber et al., 2016***). Based on the estimated susceptibility distortion, a corrected echo-planar imaging reference was calculated for a more accurate co-registration with the anatomical reference. The BOLD reference was then co-registered to the T1w reference using bbregister (FreeSurfer), which implements boundary-based registration (***Greve and Fischl, 2009***). Co-registration was configured with six degrees of freedom. The BOLD time series were resampled onto the following surfaces (FreeSurfer reconstruction nomenclature): *fsnative*. The BOLD time series (including slice-timing correction when applied) were resampled onto their original, native space by applying a single, composite transform to correct for head motion and susceptibility distortions. These resampled BOLD time series will be referred to as *preprocessed BOLD in original space*, or just *preprocessed BOLD*. The BOLD time series were resampled into standard space, generating a *preprocessed BOLD session in MNI152NLin2009cAsym space*. First, a reference volume and its skull-stripped version were generated using a custom methodology of *fMRIPrep*. Several confounding time series were calculated based on the *preprocessed BOLD*: framewise displacement (FD), DVARS, and three region-wise global signals. FD was computed using two formulations following ***Power et al., 2014*** (absolute sum of relative motions) and (***Jenkinson et al., 2002***) (relative root mean square displacement between affines). FD and DVARS were calculated for each functional session, both using their implementations in *Nipype* (following the definitions by ***Power et al., 2014***). The three global signals were extracted within the cerebrospinal fluid, white matter, and whole-brain masks.

## Mass-univariate analyses

We estimated activation differences between the different CS types for each experimental phase by computing first level contrasts of interest (e.g. CS+ > CS-), followed by a second level group analysis with a FWE correction at the cluster level. Significant cluster size was estimated in a non-parametric manner using nilearn's *non_parametric_inference* function, using 10,000 permutations. Analyses were constrained within a gray matter mask.

## RSA

We applied RSA to investigate the representational geometries of cues and contexts throughout the experiment (***Figure 1C***). Since this involved studying the neural representations during a relatively rapid event-related design, we chose to base the estimation of neural pattern similarity not on the raw BOLD data, but rather on general linear model (GLM) estimates of the trial-specific BOLD response, an approach that allows for the estimation of trial-by-trial BOLD response amplitudes, often referred to as beta-series modeling (***Rissman et al., 2004***). Specifically, we used a least square separate (LSS) approach (***Abdulrahman and Henson, 2016***), which consists of fitting one GLM for each trial, in which the tested trial is the condition of interest while controlling for all other trials. We chose this approach over the Least Square All method, which consists of fitting a single GLM, including all trials (and using each trial as a condition of interest), as LSS has been shown to be superior for dealing with collinearity, especially with fast event-related designs (***Abdulrahman and Henson, 2016***; ***Mumford et al., 2012***), which applies to our paradigm.

We estimated two separate sets of LSS models per experimental phase: one set of GLMs for CS, and a distinct set of GLMs for contexts. The onset of CS was not included in the context models, and vice versa. We reasoned that including both CS and contexts in the same LSS models was not necessary as CS type and video type were orthogonal to each other, i.e., the paradigm intrinsically controls for the potential influence of CS on context and vice versa. Furthermore, including all CS and context events in one model may risk overfitting the data (since all events but one served as variables of non-interest in each LSS model). However, to distinguish these two events temporally, the onset of the context was set as the beginning of each video and its offset as the presentation of the CS. In all models, to avoid the confounding effect of the US (electric shock) on the CS pattern, only unreinforced trials (i.e. not followed by a US) were used to conduct statistical analyses of neural pattern similarity at the group level. The onset of each US was also used as a regressor of no interest in the LSS models of the phases where US were presented (i.e. for fear acquisition and reversal). The six motion parameters (three translation and three rotation) and the average signal in the white matter and corticospinal fluid compartments were also added as regressors of non-interest. This LSS beta-series approach was implemented using Nilearn (https://nilearn.github.io/).

Trial-wise pattern similarity of each cue or context was obtained on the beta-series as the temporal correlation between all cue trials or context trials with a searchlight approach and region of interest (ROI) approach. Raw correlation values were, in both cases, Fisher r-to-z transformed before any further analyses.

## Item stability and generalization of cues

For both cues and context, we analyzed two different types of pattern similarity, i.e., item stability (within-stimulus similarity) and cue generalization (between-stimulus similarity). Item stability was defined as the average within-cue or within-context neural pattern similarity, i.e., the average neural pattern similarity between the different presentations of a given cue (out of eight possible cues per phase and in the whole experiment) or a given context (out of four possible contexts per phase, for a total of 16 contexts in the whole experiment). Item stability is defined as the average similarity of neural patterns elicited by multiple presentations of the *same* item (e.g. the kettle). It, therefore, measures the consistency of an item's neural representation across repeated encounters.

On the other hand, cue generalization was defined as the average neural pattern similarity between different exemplars of a same CS type (e.g. the similarity between the two different CS-+ items during reversal learning). Thus, while item stability provides information about the stability of the neural representation of one particular item, cue generalization expresses the formation of a higher-order association between different exemplars of the same valence category (see *Visser et al., 2013* for a similar approach). In the case of contexts, cue generalization was defined as the average neural pattern similarity between the different videos presented in a given experimental phase.

## RSA of context specificity

Specifically, for contexts, we computed the representational specificity of contexts in each experimental phase, which was defined as the difference of the average similarity of the different presentations of the same contexts (i.e. within-context similarity, or context stability) with the average similarity of the different presentations of different contexts (between-context similarity, or context generalization), separately for each experimental phase. In other words, context specificity controls for the stability of a particular stimulus (i.e. context video) with the generalization between distinct stimuli (all other contexts shown in an experimental phase), with higher context specificity entailing more distinct representations of contexts in each experimental phase. We then compared the context specificity maps between phases in order to assess the effect of experimental manipulation (acquisition, reversal, and test phases) on context specificity.

## Searchlight approach

The different types of similarity analyses described above were implemented in a searchlight approach. Pattern similarity was estimated at the voxel level using the searchlight algorithm as implemented in brainIAK (https://brainiak.org/). A square with a radius of twice the voxel size (i.e. 5 mm radius) was used with each brain voxel as the center to estimate the average pattern similarity within the searchlight. Voxels were included only if at least 50% of their surrounding voxels were included in the brain

mask. A pattern similarity formula specific to each type of pattern similarity (e.g. item stability and cue generalization for CS or contexts) was used. Hypothesis testing was performed by comparing the obtained Fisher r-to-z-transformed correlation maps of the conditions of interest (e.g. CS+ cue generalization vs. CS- cue generalization). Significance corresponding to the contrast between conditions of the maps of interest was FWER-corrected using non-parametric permutation testing at the cluster level (10,000 permutations) to estimate significant cluster size. Additionally, we adjusted the alpha threshold against which we assessed the significance of the cluster-specific FWER-corrected p-values using Bonferroni correction. In this order, we divided the default alpha-corrected threshold of 0.05 by the number of statistical comparisons that were conducted in each experimental phase. For example, for fear acquisition, we compared the CS+ > CS- contrast for both item stability and cue generalization, resulting in 2 comparisons and hence a corrected alpha threshold of 0.025. Only clusters that had a FWER-corrected p-value below the Bonferroni-adjusted threshold were deemed significant. All searchlight analyses were restricted within a gray matter mask.

### ROI-based RSA

Additionally, pattern similarity analyses were performed at the ROI level, using the results from the searchlight analyses to carry out additional ROI-based exploratory analyses. We thresholded the statistical maps of the searchlight analyses to only retain (corrected) significant clusters. We used nilearn's *connected_region* function to define individual ROIs statistical maps, using a minimum ROI size of 1500 mm$^3$. These ROI masks in MNI space were used to estimate the average neural pattern similarity within each ROI, defined as the correlation between the BOLD response of all trials. Raw correlation values were then Fisher r-to-z transformed. The effects of experimental phase and type of neural pattern similarity (within/between) were assessed with LME models using the *lme4* package in R. Significance was assessed with the Satterthwaite method for estimating degrees of freedom using maximum likelihood. ROI-based analyses were corrected for multiple comparisons using False Discovery Rate (FDR) for all ROI-specific comparisons given the exploratory character of these analyses.

## Acknowledgements

This work was supported by the Deutsche Forschungsgemeinschaft (DFG, German Research Foundation) through grant SFB 1280 (projects A02 and A19) project number 316803389.

## Additional information

### Funding

| Funder | Grant reference number | Author |
|---|---|---|
| Deutsche Forschungsgemeinschaft | 316803389 | Antoine Bouyeure<br>Daniel Pacheco-Estefan<br>Marie-Christin Fellner<br>George Jacob<br>Malte Kobelt<br>Jonas Rose<br>Nikolai Axmacher |
| Spanish Ministry of Science and Innovation | 2024B0303390003 | Daniel Pacheco-Estefan |

The funders had no role in study design, data collection and interpretation, or the decision to submit the work for publication.

### Author contributions

Antoine Bouyeure, Conceptualization, Resources, Data curation, Software, Formal analysis, Validation, Investigation, Visualization, Methodology, Writing – original draft, Writing – review and editing; Daniel Pacheco-Estefan, Conceptualization, Formal analysis, Validation, Investigation, Methodology, Writing – review and editing; George Jacob, Conceptualization, Investigation, Methodology, Writing – original draft, Writing – review and editing; Malte Kobelt, Investigation, Methodology, Writing – review and editing; Marie-Christin Fellner, Conceptualization, Data curation, Software, Investigation,

Methodology; Jonas Rose, Conceptualization, Validation, Investigation, Writing – review and editing; Nikolai Axmacher, Conceptualization, Formal analysis, Supervision, Funding acquisition, Validation, Investigation, Methodology, Writing – original draft, Project administration, Writing – review and editing

### Author ORCIDs
Antoine Bouyeure [ID] https://orcid.org/0000-0002-0689-6878

### Ethics
All participants provided written informed consent before participation and were unaware of the aims of the experiment. The procedures were performed in accordance with the tenets of the Declaration of Helsinki and were approved by the ethical review board of the Faculty of Psychology at Ruhr University Bochum.

Reviewer #1 (Public review): https://doi.org/10.7554/eLife.105126.3.sa1
Reviewer #2 (Public review): https://doi.org/10.7554/eLife.105126.3.sa2
Author response https://doi.org/10.7554/eLife.105126.3.sa3

## Additional files

### Supplementary files
Supplementary file 1. Wilcoxon post-hoc tests and cluster statistics. (a) Wilcoxon post-hoc tests, with W and corrected p-values, for the pairwise comparisons of unconditioned stimulus (US) expectancy between each conditioned stimulus (CS) type for each learning phase. (b) Cluster statistics for the cue generalization (CS+ > CS−) contrast in fear acquisition. FWER-corrected p-values of the clusters are marked as significant according to a Bonferroni-corrected threshold of $p<0.025$ (0.05/2). (c) Cluster statistics for cue generalization and item stability in fear reversal. FWER-corrected p-values of the clusters are marked as significant according to a Bonferroni-corrected threshold of $p<0.00625$ (0.05/8). (d) Cluster statistics for item stability in test new and test old. FWER-corrected p-values of the clusters are marked as significant according to a Bonferroni-corrected threshold of $p<0.00625$ (0.05/8). (e) Cluster statistics for context specificity differences between acquisition and reversal. P-values are based on a two-tailed paired t-test with 24 participants. Clusters are marked as significant according to a Bonferroni-corrected threshold of $p<0.00714$ (0.05/7).

MDAR checklist

### Data availability
Group-level Statistical Maps: Statistical maps for fMRI contrasts and searchlight-RSA analyses have been deposited in NeuroVault (https://identifiers.org/neurovault.collection:23032) Source Data: The individual-level data used to generate figures, including mean US expectancy ratings, individual LSS maps, individual RSA searchlight maps, for each participant and experimental phase, are available as Source Data files on the Open Science Framework (OSF) (https://doi.org/10.17605/OSF.IO/NGWKA) Analysis Code: Custom scripts used for behavioral analysis, fMRI preprocessing, and representational similarity analysis (R/Python/MATLAB) are publicly available on GitHub (https://github.com/AntoineBouyeure/Representational-properties-of-cues-and-contexts-shape-fear-learning-and-reversal copy archived at *Bouyeure, 2026*).Behavioral paradigm: Scripts used for the behavioral paradigm are available on GitHub (https://github.com/AntoineBouyeure/Representational-properties-of-cues-and-contexts-shape-fear-learning-and-reversal).

The following datasets were generated:

| Author(s) | Year | Dataset title | Dataset URL | Database and Identifier |
|---|---|---|---|---|
| Bouyeure A | 2026 | Distinct representational properties of cues and contexts shape fear learning and extinction | https://doi.org/10.17605/OSF.IO/NGWKA | Open Science Framework, 10.17605/OSF.IO/NGWKA |
| Bouyeure A | 2026 | Distinct representational properties of cues and contexts shape fear and reversal learning | https://identifiers.org/neurovault.collection:23032 | NeuroVault, 23032 |

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
