## [Editor Report · eLife Assessment]

This is an **important** study with **convincing** evidence that multi-voxel fMRI activity patterns for threat-conditioned stimuli are altered by learning CS-US contingencies. The analyses are dense, but rigorous. The protocol is quite nuanced and complex, but the authors have done a fair job of explaining and presenting the results. The work is relevant for our understanding of how effective learning changes neural stimulus representation in the human brain.

---

## [Referee Report · Reviewer #1 (Public review)]

Summary:

The authors conducted a human neuroimaging study investigating the role of context in the representation of fear associations when the contingencies between a conditioned stimulus and shock unconditioned stimulus switches between contexts. The novelty of the analysis centered on neural pattern similarity to derive a measure of context and cue stability and generalization across different regions of the brain. Given the complexity and nuance of the results, it is kind of difficult to provide a concise summary. But during fear and reversal, there was cue generalization (between current CS+ cues) in the canonical fear network, and "item stability" for cues that changed their association with the shock in the IFG and precuneus. Reinstatement was quantified as pattern similarity for items or sets of cues from the earlier phases to the test phases, and they found different patterns in the IFG and dmPFC. A similar analytical strategy was applied to contexts.

Strengths:

Overall, I found this to be a novel use of MVPA to study the role of context in reversal/extinction of human fear conditioning that yielded interesting results. The paper was overall well-written, with a strong introduction and fairly detailed methods and results. The lack of any univariate contrast results from the test phases was used as motivation for the neural pattern similarity approach, which I appreciated as a reader.

I have no additional or new comments. The authors adequately addressed my major comments and concerns.

---

## [Referee Report · Reviewer #2 (Public review)]

Summary:

This is a timely and original study on the geometry of macroscopic (2.5 mm) brain representations of multiple cues and contexts in Pavlovian fear conditioning. The authors report that these representations differ between initial learning, and reversal learning, and remain stable during extinction.

Strengths:

The authors address an important question and use a rigorous experimental methodology.

Weaknesses:

The findings are limited by the chosen spatial resolution (2.5 mm) which is far away from what modern fMRI can achieve. Also, region-of-interesting findings should be considered exploratory due to the chosen FDR method for correction for multiple comparison (which is transparently reported).

---

## [Author Response]

The following is the authors’ response to the original reviews.

**Reviewing Editor Comments:**
The study design used reversal learning (i.e. the CS+ becomes the CS- and vice versa), while the title mentions 'fear learning and extinction'. In my opinion, the paper does not provide insight into extinction and the title should be changed.

Thank you for this important point. We agree that our paradigm focuses more directly on reversal learning than on standard extinction, as the test phases represent extinction in the absence of a US but follow a reversal phase. To better reflect the core of our investigation, we have changed the title.

Proposed change in manuscript (Title): Original Title: Distinct representational properties of cues and contexts shape fear learning and extinction

New Title: Distinct representational properties of cues and contexts shape fear and reversal learning

Secondly, the design uses 'trace conditioning', whereas the neuroscientific research and synaptic/memory models are rather based on 'delay conditioning'. However, given the limitations of this design, it would still be possible to make the implications of this paper relevant to other areas, such as declarative memory research.

This is an excellent point, and we thank you for highlighting it. Our design, where a temporal gap exists between the CS offset and US onset, is indeed a form of trace conditioning. We also agree that this feature, particularly given the known role of the hippocampus in trace conditioning, strengthens the link between our findings and the broader field of episodic memory.

Proposed change in manuscript (Methods, Section "General procedure and stimuli"): We inserted the following text (lines 218-220): "It is important to note that the temporal gap between the CS offset and potential US delivery (see Figure 1A) indicates that our paradigm employs a trace conditioning design. This form of learning is known to be hippocampus-dependent and has been distinguished from delay conditioning.

Proposed change in manuscript (Discussion): We added the following to the discussion (lines 774-779): "Furthermore, our use of a trace conditioning paradigm, which is known to engage the hippocampus more than delay conditioning does, may have facilitated the detection of item-specific, episodiclike memory traces and their interaction with context. This strengthens the relevance of our findings for understanding the interplay between aversive learning and mechanisms of episodic memory."

The strength of the evidence at this point would be described as 'solid'. In order to increase the strength (to convincing), analyses including FWE correction would be necessary. I think exploratory (and perhaps some FDR-based) analyses have their valued place in papers, but I agree that these should be reported as such. The issue of testing multiple independent hypotheses also needs to be addressed to increase the strength of evidence (to convincing). Evaluating the design with 4 cues could lead to false positives if, for example, current valence, i.e. (CS++ and CS-+) > (CS+- and CS--), and past valence (CS++ > CS+-) > (CS-+ > CS--) are tested as independent tests within the same data set. Authors need to adjust their alpha threshold.

We fully agree. As summarized in our general response, we have implemented two major changes to our statistical approach to address these concerns comprehensively. These, are stated above, are the following:

(1) Correction for Multiple Hypotheses: We previously used FWER-corrected p-values that were obtained through permutation testing. We have now applied a Bonferroni adjustment to the FWER-corrected threshold (previously 0.05) used in our searchlight analyses. For instance, in the acquisition phase, since 2 independent tests (contrasts) were conducted, the significance threshold of each of these searchlight maps was set to p <0.025 (after FWE-correction estimated through non-parametric permutation testing); in reversal, 4 tests were conducted, hence the significance threshold was set to p<0.0125. This change is now clearly described in the Methods section (section “Searchlight approach” lines 477484). This change had no impact on our searchlight results, given that all clusters that were previously as significant with the previous FWER alpha of 0.05 were also significant at the new, Bonferroni-adjusted thresholds; we also now report the cluster-specific corrected p-values in the cluster tables in Supplementary Material.**(**2) ROI Analyses: Our ROI-based analyses used FDR-based correction for within each item reinstatement/generalized reinstatement pair of each ROI. We now explicitly state in the abstract, methods and results sections that these ROI-based analyses are exploratory and secondary to the primary whole-brain results, given that the correction method used is more liberal, in accordance with the exploratory character of these analyses.

We are confident that these changes ensure both the robustness and transparency of our reported findings.

**Reviewer #1 (Public Review):**
(1) I had a difficult time unpacking lines 419-420: "item stability represents the similarity of the neural representation of an item to other representations of this same item."

We thank the reviewer for pointing out this lack of clarity. We have revised the definition to be more intuitive and have ensured it is introduced earlier in the manuscript.

Proposed change in manuscript (Introduction, lines 144-150): We introduced the concept earlier and more clearly: "Furthermore, we can measure the consistency of a neural pattern for a given item across multiple presentations. This metric, which we refer to as “item stability”, quantifies how consistently a specific stimulus (e.g., the image of a kettle) is represented in the brain across multiple repetitions of the same item. Higher item stability has been linked to successful episodic memory encoding (Xue et al., 2010)."

Proposed change in manuscript (Methods, Section "Item stability and generalization of cues"): Original text: "Thus, item stability represents the similarity of the neural representation of an item to other representations of this same item (Xue, 2018), or the consistency of neural activity across repetitions (Sommer et al., 2022)."

Revised text (lines 434-436): "Item stability is defined as the average similarity of neural patterns elicited by multiple presentations of the same item (e.g., the kettle). It therefore measures the consistency of an item's neural representation across repeated encounters."

(2) The authors use the phrase "representational geometry" several times in the paper without clearly defining what they mean by this.

We apologize for this omission. We have now added a clear and concise definition of "representational geometry" in the Introduction, citing the foundational work by Kriegeskorte et al. (2008).

Proposed change in manuscript (Introduction): We inserted the following text (lines 117-125): " By contrast, multivariate pattern analyses (MVPA), such as representational similarity analysis (RSA; Kriegeskorte et al., 2008) has emerged as a powerful tool to investigate the content and structure of these representations (e.g., Hennings et al., 2022). This approach allows us to characterize the “representational geometry” of a set of items – that is, the structure of similarities and dissimilarities between their associated neural activity patterns. This geometry reveals how the brain organizes information, for instance, by clustering items that are conceptually similar while separating those that are distinct."

(3) The abstract is quite dense and will likely be challenging to decipher for those without a specialized knowledge of both the topic (fear conditioning) and the analytical approach. For instance, the goal of the study is clearly articulated in the first few sentences, but then suddenly jumps to a sentence stating "our data show that contingency changes during reversal induce memory traces with distinct representational geometries characterized by stable activity patterns across repetitions..." this would be challenging for a reader to grok without having a clear understanding of the complex analytical approach used in the paper.

We agree with your assessment. We have rewritten it to be more accessible to a general scientific audience, by focusing on the conceptual findings rather than methodological jargon.

Proposed change in manuscript (Abstract): We revised the abstract to be clearer. It now reads: " When we learn that something is dangerous, a fear memory is formed. However, this memory is not fixed and can be updated through new experiences, such as learning that the threat is no longer present. This process of updating, known as extinction or reversal learning, is highly dependent on the context in which it occurs. How the brain represents cues, contexts, and their changing threat value remains a major question. Here, we used functional magnetic resonance imaging and a novel fear learning paradigm to track the neural representations of stimuli across fear acquisition, reversal, and test phases. We found that initial fear learning creates generalized neural representations for all threatening cues in the brain’s fear network. During reversal learning, when threat contingencies switched for some of the cues, two distinct representational strategies were observed. On the one hand, we still identified generalized patterns for currently threatening cues, whereas on the other hand, we observed highly stable representations of individual cues (i.e., item-specific) that changed their valence, particularly in the precuneus and prefrontal cortex. Furthermore, we observed that the brain represents contexts more distinctly during reversal learning. Furthermore, additional exploratory analyses showed that the degree of this context specificity in the prefrontal cortex predicted the subsequent return of fear, providing a potential neural mechanism for fear renewal. Our findings reveal that the brain uses a flexible combination of generalized and specific representations to adapt to a changing world, shedding new light on the mechanisms that support cognitive flexibility and the treatment of anxiety disorders via exposure therapy."

(4) Minor: I believe it is STM200 not the STM2000.

Thank you for pointing this out. We have corrected it in the Methods section.

Proposed change in manuscript (Methods, Page 5, Line 211): Original: STM2000 -> Corrected: STM200

(5) Line 146: "...could be particularly fruitful as a means to study the influence of fear reversal or extinction on context representations, which have never been analyzed in previous fear and extinction learning studies." I direct the authors to Hennings et al., 2020, Contextual reinstatement promotes extinction generalization in healthy adults but not PTSD, as an example of using MVPA to decipher reinstatement of the extinction context during test.

Thank for pointing us towards this relevant work. We have revised the sentence to reflect the state of the literature more accurately.

Proposed change in manuscript (Introduction, Page 3): Original text: "...which have never been analyzed in previous fear and extinction learning studies."

Revised text (lines 154-157): "...which, despite some notable exceptions (e.g., Hennings et al., 2020), have been less systematically investigated than cue representations across different learning stages."

(6) This is a methodological/conceptual point, but it appears from Figure 1 that the shock occurs 2.5 seconds after the CS (and context) goes off the screen. This would seem to be more like a trace conditioning procedure than a standard delay fear conditioning procedure. This could be a trivial point, but there have been numerous studies over the last several decades comparing differences between these two forms of fear acquisition, both behaviorally and neurally, including differences in how trace vs delay conditioning is extinguished.

Thank you for this pertinent observation; this was also pointed out by the editor. As detailed in our response to the editor, we now explicitly acknowledge that our paradigm uses a trace conditioning design, and have added statements to this effect in the Methods and Discussion sections (lines 218-220, and 774-779).

(7) In Figure 4, it would help to see the individual data points derived from the model used to test significance between the different conditions (reinstatement between Acq, reversal, and test-new).

We agree that this would improve the transparency of our results. We have revised Figure 4 to include individual data points, which are now plotted over the bar graphs.

**Reviewer #2 (Public Review & Recommendations)**
Use a more stringent method of multiple comparison correction: voxel-wise FWE instead of FDR; Holm-Bonferroni across multiple hypothesis tests. If FDR is chosen then the exploratory character of the results should be transparently reported in the abstract.

Thank you for these critical comments regarding our statistical methods. As detailed in the general response and response to the editor (Comment 3), we have thoroughly revised our approach to ensure its rigor. We now clarify that our whole-brain analyses consistently use FWER-corrected pvalues. Additionally, the significance of these FWER-corrected p-values (obtained through permutation testing), which were previously considered significant against a default threshold of 0.05, are now compared with a Bonferroni-adjusted threshold equal to the number of tested contrasts in each experimental phase. We have modified the revised manuscript accordingly, in the methods section (lines 473-484) and in the supplementary material, where we added the p-values (FWER-corrected) of each cluster, evaluated against the new Bonferroni-adjusted thresholds. It is to be of note that this had no impact on our searchlight results, given that all clusters that were previously reported as significant with the alpha threshold of 0.05 were also significant at the new, corrected thresholds.

Proposed change in manuscript (Methods): We revised the relevant paragraphs (lines 473-484): "Significance corresponding to the contrast between conditions of the maps of interest was FWER-corrected using nonparametric permutation testing at the cluster level (10,000 permutations) to estimate significant cluster size. Additionally, we adjusted the alpha threshold against which we assessed the significance of the cluster-specific FWERcorrected p-values using Bonferroni correction. In this order, we divided the default alpha corrected threshold of 0.05 by the number of statistical comparisons that were conducted in each experimental phase. For example, for fear acquisition, we compared the CS+>CS- contrast for both item stability and cue generalization, resulting in 2 comparisons and hence a corrected alpha threshold of 0.025. Only clusters that had a FWER-corrected p-value below the Bonferroni-adjusted threshold were deemed significant. All searchlight analyses were restricted within a gray matter mask.”

The authors report fMRI results from line 96 onwards; all of these refer exclusively to mass-univariate fMRI which could be mentioned more transparently... The authors contrast "activation fMRI" with "RSA" (line 112). Again, I would suggest mentioning "mass-univariate fMRI", and contrasting this with "multivariate" fMRI, of which RSA is just one flavour. For example, there is some work that is clear and replicable, demonstrating human amygdala involvement in fear conditioning using SVM-based analysis of highresolution amygdala signals (one paper is currently cited in the discussion).

Thank you for this important clarification. We have revised the manuscript to incorporate your suggestions. We now introduce our initial analyses as "mass-univariate" and contrast them with the "multivariate pattern analysis" (MVPA) approach of RSA.

Proposed change in manuscript (Introduction): We revised the relevant paragraphs (lines 113-125): " While mass-univariate functional magnetic resonance imaging (fMRI) activation studies have been instrumental in identifying the brain regions involved in fear learning and extinction, they are insensitive to the patterns of neural activity that underlie the stimulus-specific representations of threat cues and contexts. Contrastingly, multivariate pattern analyses methods, such as representational similarity analysis (RSA; Kriegeskorte et al., 2008), have emerged as a powerful tool to investigate the content and structure of these representations (e.g., Hennings et al., 2022). This approach allows us to characterize the “representational geometry” of a set of items – i.e., the structure of similarities and dissimilarities between their associated neural activity patterns. This geometry reveals how the brain organizes information, for instance, by clustering items that are conceptually similar while separating those that are distinct.”

Line 177: unclear how incomplete data was dealt with. If there are 30 subjects and 9 incomplete data sets, then how do they end up with 24 in the final sample?

We apologize for the unclear wording in our original manuscript. We have clarified the participant exclusion pipeline in the Methods section.

Proposed change in manuscript (Methods, Section "Participants"): Original text: "The number of participants with usable fMRI data for each phase was as follows: N = 30 for the first phase of day one, N = 29 for the second phase of day one, N = 27 for the first phase of day two, and N = 26 for the second phase of day two. Of the 30 participants who completed the first session, four did not return for the second day and thus had incomplete data across the four experimental phases. An additional two participants were excluded from the analysis due to excessive head movement (>2.5 mm in any direction). This resulted in a final sample of 24 participants (8 males) between 18 and 32 years of age (mean: 24.69 years, standard deviation: 3.6) with complete, low-motion fMRI data for all analyses."

Revised text: "The number of participants with usable fMRI data for each phase was as follows: N = 30 for the first phase of day one, N = 29 for the second phase of day one, N = 27 for the first phase of day two, and N = 26 for the second phase of day two. An additional two participants were excluded from the analysis due to excessive head movement (>2.5 mm in any direction). This resulted in a final sample of 24 participants (8 males) between 18 and 32 years of age (mean: 24.69 years, standard deviation: 3.6) with complete, low-motion fMRI data for all analyses."

Typo in line 201.

Thank you for your comment. We have re-examined line 201 (“interval (Figure 1A). A total of eight CSs were presented during each phase and”) and the surrounding text but were unable to identify a clear typographical error in the provided quote. However, in the process of revising the manuscript for clarity, we have rephrased this section.

it would be good to see all details of the US calibration procedure, and the physical details of the electric shock (e.g. duration, ...).

Thank you for your comment. We have expanded the Methods section to include these important details.

Proposed change in manuscript (Methods, Section "General procedure and stimuli"): We inserted the following text (lines 225-230): "Electrical stimulation was delivered via two Ag/AgCl electrodes attached to the distal phalanx of the index and middle fingers of the non-dominant hand. he intensity of the electrical stimulation was calibrated individually for each participant prior to the experiment. Using a stepping procedure, the voltage was gradually increased until the participant rated the sensation as 'unpleasant but not painful'.

"beta series modelling" is a jargon term used in some neuroimaging software but not others. In essence, the authors use trial-by-trial BOLD response amplitude estimates in their model. Also, I don't think this requires justification - using the raw BOLD signal would seem outdated for at least 15 years.

Thank you for this helpful suggestion. We have simplified the relevant sentences for improved clarity.

Proposed change in manuscript (Methods, Section "RSA"): Original text: "...an approach known as beta-series modeling (Rissman et al., 2004; Turner et al., 2012)."

Revised text (lines 391-393): "...an approach that allows for the estimation of trial-by-trial BOLD response amplitudes, often referred to as beta-series modeling (Rissman et al., 2004). Specifically, we used a Least Square Separate (LSS) approach..."

I found the use of "Pavlovian trace" a bit confusing. The authors are coming from memory research where "memory trace" is often used; however, in associative learning the term "trace conditioning" means something else. Perhaps this can be explained upon first occurrence, and "memory trace" instead of "Pavlovian trace" might be more common.

We are grateful for this comment, as it highlights a critical point of potential confusion, especially given that we now acknowledge our paradigm uses a trace conditioning design. To eliminate this ambiguity, we have replaced all instances of "Pavlovian trace" with "lingering fear memory trace" throughout the manuscript (lines 542 and 599).

I would suggest removing evaluative statements from the results (repeated use of "interesting").

Thank you for this valuable suggestion. We have reviewed the Results section and removed subjective evaluative words to maintain a more objective tone.

Line 882: one of these references refers to a multivariate BOLD analysis using SVM, not explicitly using temporal information in the signal (although they do show session-by-session information).

Thank you for this correction. We have re-examined the cited paper (Bach et al., 2011) and removed its inclusion in the text accordingly.